# Mixture Matrix Completion

**Daniel Pimentel-Alarcón**
Department of Computer Science
Georgia State University
Atlanta, GA, 30303
pimentel@gsu.edu

## Abstract

Completing a data matrix $\mathbf{X}$ has become an ubiquitous problem in modern data science, with motivations in recommender systems, computer vision, and networks inference, to name a few. One typical assumption is that $\mathbf{X}$ is low-rank. A more general model assumes that each *column* of $\mathbf{X}$ corresponds to one of several low-rank matrices. This paper generalizes these models to what we call *mixture matrix completion* (MMC): the case where each *entry* of $\mathbf{X}$ corresponds to one of several low-rank matrices. MMC is a more accurate model for recommender systems, and brings more flexibility to other completion and clustering problems. We make four fundamental contributions about this new model. First, we show that MMC is theoretically possible (well-posed). Second, we give its precise information-theoretic identifiability conditions. Third, we derive the sample complexity of MMC. Finally, we give a practical algorithm for MMC with performance comparable to the state-of-the-art for simpler related problems, both on synthetic and real data.

## 1 Introduction

Matrix completion aims to estimate the missing entries of an incomplete data matrix $\mathbf{X}$. One of its main motivations arises in recommender systems, where each row represents an item, and each column represents a user. We only observe an entry in $\mathbf{X}$ whenever a user rates an item, and the goal is to predict unseen ratings in order to make good recommendations.

**Related Work.** In 2009, Candès and Recht [1] introduced *low-rank matrix completion* (LRMC), arguably the most popular model for this task. LRMC assumes that each column (user) can be represented as a linear combination of a few others, whence $\mathbf{X}$ is low-rank. Later in 2012, Eriksson et. al. [2] introduced *high-rank matrix completion* (HRMC), also known as *subspace clustering with missing data*. This more general model assumes that each column of $\mathbf{X}$ comes from one of several low-rank matrices, thus allowing several types of users. Since their inceptions, both LRMC and HRMC have attracted a tremendous amount of attention (see [1–27] for a very incomplete list).

**Paper contributions.** This paper introduces an even more general model: *mixture matrix completion* (MMC), which assumes that each *entry* in $\mathbf{X}$ (rather than column) comes from one out of several low-rank matrices, and the goal is to recover the matrices in the mixture. Figure 1 illustrates the generalization from LRMC to HRMC and to MMC. One of the main motivations behind MMC is that users often share the same account, and so each column in $\mathbf{X}$ may contain ratings from several users. Nonetheless, as we show in Section 2, MMC is also a more accurate model for many other contemporary applications, including networks inference, computer vision, and metagenomics. This paper makes several fundamental contributions about MMC:

- **Well posedness.** First, we show that MMC is theoretically possible if we observe *the right entries* and the mixture is *generic* (precise definitions below).

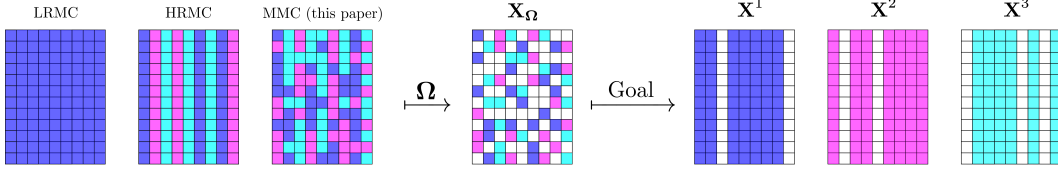

Figure 1: In LRMC, $\mathbf{X}$ is a low-rank matrix. In HRMC, each *column* of $\mathbf{X}$ comes from one of several low-rank matrices. In MMC, each *entry* comes from one of several low-rank matrices $\mathbf{X}^1, \ldots, \mathbf{X}^K$; we only observe $\mathbf{X}_{\mathbf{\Omega}}$, and our goal is to recover the columns of $\mathbf{X}^1, \ldots, \mathbf{X}^K$ that have observations in $\mathbf{X}_{\mathbf{\Omega}}$.

- **Identifiability conditions.** We provide precise information-theoretical conditions on the entries that need to be observed such that a mixture of $K$ low-rank matrices is identifiable. These extend similar recent results of LRMC [3] and HRMC [4] to the setting of MMC. The subtlety in proving these results is that there could exist *false* mixtures that agree with the observed entries, even if the sampling is uniquely completable for LRMC and HRMC (see Example 1). In other words, there exits samplings that are identifiable for LRMC (and HRMC) but are not identifiable for MMC, and so in general it is not enough to simply have $K$ times more samples. Hence, it was necessary to derive identifiability conditions for MMC, similar to those of LRMC in [3] and HRMC in [4]. We point out that in contrast to typical completion theory [1, 2, 5–20], these type of identifiability conditions are deterministic (not restricted to uniform sampling), and make no coherence assumptions.

- **Sample complexity.** If $\mathbf{X} \in \mathbb{R}^{d \times n}$ is a mixture of $K$ rank-$r$ matrices, we show that with high probability, our identifiability conditions will be met if each entry is observed with probability $\mathcal{O}(\frac{K}{d} \max\{r, \log d\})$, thus deriving the sample complexity of MMC, which is the same as the sample complexity of HRMC [4], and simplifies to $\mathcal{O}(\frac{1}{d} \max\{r, \log d\})$ in the case of $K = 1$, which corresponds to the sample complexity of LRMC [3]. Intuitively, this means that information-theoretically, we virtually pay no price for mixing low-rank matrices.

- **Practical algorithm.** Our identifiability results follow from a combinatorial analysis that is infeasible in practice. To address this, we give a practical alternating algorithm for MMC whose performance (in the more difficult problem of MMC) is comparable to state-of-the-art algorithms for the much simpler problems of HRMC and LRMC.

## 2  Motivating Applications

Besides recommender systems, there are many important applications where data can be modeled as a mixture of low-rank matrices. Here are a few examples motivated by current data science challenges.

**Networks Inference.** Estimating the topology of a network (internet, sensor networks, biological networks, social networks) has been the subject of a large body of research in recent years [28–34]. To this end, companies routinely collect distances between nodes (e.g., computers) that connect with monitors (e.g., Google, Amazon, Facebook) in a data matrix $\mathbf{X}$. In a simplified model, if node $j$ is in subnet $k$, then the $j^{\text{th}}$ column can be modeled as the sum of (i) the distance between node $j$ and router $k$, and (ii) the distance between router $k$ and each of the monitors. Hence, the columns (nodes) corresponding to each subnet form a low-rank matrix, which is precisely the model assumed by HRMC. However, depending on the network's traffic, each node may use different routes to communicate at different times. Consequently, the same column in $\mathbf{X}$ may contain measurements from different low-rank matrices. In other words, distance matrices of networks are a mixture of low-rank matrices.

**Computer Vision.** Background segmentation is one of the most fundamental and crucial tasks in computer vision, yet it can be tremendously challenging. The vectorized frames of a video can be modeled as columns with some entries (pixels) in a low-rank background, and some outlier entries, corresponding to the foreground. Typical methods, like the acclaimed Robust PCA (principal component analysis) [35–46], assume that the foreground is sparse and has no particular structure. However, in many situations this is not the case. For instance, since the location of an object in consecutive frames is highly correlated, the foreground can be highly structured. Similarly, the foreground may not be sparse, specially if there are foreground objects moving close to the camera

(e.g., in a selfie). Even state-of-the-art methods fail in scenarios like these, which are not covered by current models (see Figure 3 for an example). In contrast, MMC allows to use one matrix in the mixture to represent the background, other matrices to represent foreground objects (small or large, even dominant), and even other matrices to account for occlusions and other illumination/visual artifacts. Hence, MMC can be a more accurate model for video segmentation and other image processing tasks, including inpainting [47] and face clustering, which we explore in our experiments.

**Metagenomics.** One contemporary challenge in Biology is to quantify the presence of different types of bacteria in a system (e.g., the human gut microbiome) [48–52]. The main idea is to collect several DNA samples from such a system, and use their genomic information to count the number of bacteria of each type (the genome of each bacterium determines its type). In practice, to obtain an organism's genome (e.g., a person's genome), biologists feed a DNA sample (e.g., blood or hair) to a sequencer machine that produces a series of *reads*, which are short genomic sequences that can later be assembled and aligned to recover the entire genome. The challenge arises when the sequencer is provided a sample with DNA from multiple organisms, as is the case in the human gut microbiome, where any sample will contain a mixture of DNA from multiple bacteria that cannot be disentangled into individual bacterium. In this case, each read produced by the sequencer may correspond to a different type of bacteria. Consequently, each DNA sample (column) may contain genes (rows) from different types of bacteria, which is precisely the model that MMC describes.

## 3  Problem Statement

Let $\mathbf{X}^1, \ldots, \mathbf{X}^K \in \mathbb{R}^{d \times n}$ be a set of rank-$r$ matrices, and let $\mathbf{\Omega}^1, \ldots, \mathbf{\Omega}^k \in \{0,1\}^{d \times n}$ indicate *disjoint* sets of observed entries. Suppose $\mathbf{X}^1, \ldots, \mathbf{X}^K$ and $\mathbf{\Omega}^1, \ldots, \mathbf{\Omega}^K$ are unknown, and we only observe $\mathbf{X}_\mathbf{\Omega}$, defined as follows:

- If the $(i, j)^{th}$ entry of $\mathbf{\Omega}^k$ is 1, then the $(i, j)^{th}$ entry of $\mathbf{X}_\mathbf{\Omega}$ is equal to the $(i, j)^{th}$ entry of $\mathbf{X}^k$.

- If the $(i, j)^{th}$ entry of $\mathbf{\Omega}^k$ is 0 for every $k = 1, \ldots, K$, then the $(i, j)^{th}$ entry of $\mathbf{X}_\mathbf{\Omega}$ is missing.

This way $\mathbf{\Omega}^k$ indicates the entries of $\mathbf{X}_\mathbf{\Omega}$ that correspond to $\mathbf{X}^k$, and $\mathbf{\Omega} := \sum_{k=1}^{K} \mathbf{\Omega}^k$ indicates the set of *all* observed entries. Since $\mathbf{\Omega}^1, \ldots, \mathbf{\Omega}^K$ are disjoint, $\mathbf{\Omega} \in \{0,1\}^{d \times n}$. Equivalently, each observed entry of $\mathbf{X}_\mathbf{\Omega}$ corresponds to an entry in either $\mathbf{X}^1$ or $\mathbf{X}^2$ or ... or $\mathbf{X}^K$ (i.e., there are no *collisions*). In words, $\mathbf{X}_\mathbf{\Omega}$ contains a *mixture* of entries from several low-rank matrices.

The goal of MMC is to recover all the columns of $\mathbf{X}^1, \ldots, \mathbf{X}^K$ that have observations in $\mathbf{X}_\mathbf{\Omega}$ (see Figure 1 to build some intuition). In our recommendations example, a column $\mathbf{x}_\omega \in \mathbf{X}_\mathbf{\Omega}$ will contain entries from $\mathbf{X}^k$ whenever $\mathbf{x}_\omega$ contains ratings from a user of the $k^{th}$ type. Similarly, the same column will contain entries from $\mathbf{X}^\ell$ whenever it also contains ratings from a user of the $\ell^{th}$ type. We would like to predict the preferences of both users, or more generally, all users that have ratings in $\mathbf{x}_\omega$. On the other hand, if $\mathbf{x}_\omega$ has no entries from $\mathbf{X}^k$, then $\mathbf{x}_\omega$ involves no users of the $k^{th}$ type, and so it would be impossible (and futile) to try to recover such column of $\mathbf{X}^k$. In MMC, the matrices $\mathbf{\Omega}^1, \ldots, \mathbf{\Omega}^K$ play the role of the *hidden* variables constantly present in mixture problems. Notice that if we knew $\mathbf{\Omega}^1, \ldots, \mathbf{\Omega}^K$, then we could partition $\mathbf{X}_\mathbf{\Omega}$ accordingly, and estimate $\mathbf{X}^1, \ldots, \mathbf{X}^K$ using standard LRMC. The challenge is that we do not know $\mathbf{\Omega}^1, \ldots, \mathbf{\Omega}^K$.

### 3.1  The Subtleties of MMC

The main theoretical difficulty of MMC is that depending on the pattern of missing data, there could exist *false* mixtures. That is, matrices $\tilde{\mathbf{X}}^1, \ldots, \tilde{\mathbf{X}}^K$, other than $\mathbf{X}^1, \ldots, \mathbf{X}^K$, that agree with $\mathbf{X}_\mathbf{\Omega}$, even if $\mathbf{X}^1, \ldots, \mathbf{X}^K$ are observed on uniquely completable patterns for LRMC.

**Example 1.** *Consider the next rank-1 matrices $\mathbf{X}^1, \mathbf{X}^2$, and their partially observed mixture $\mathbf{X}_\mathbf{\Omega}$:*

$$\mathbf{X}^1 = \begin{bmatrix} 1 & 2 & 3 & 4 \\ 1 & 2 & 3 & 4 \\ 1 & 2 & 3 & 4 \\ 1 & 2 & 3 & 4 \\ 1 & 2 & 3 & 4 \end{bmatrix}, \quad \mathbf{X}^2 = \begin{bmatrix} 1 & 2 & 3 & 4 \\ 2 & 4 & 6 & 8 \\ 3 & 6 & 9 & 12 \\ 4 & 8 & 12 & 16 \\ 5 & 10 & 15 & 20 \end{bmatrix}, \quad \mathbf{X}_\mathbf{\Omega} = \begin{bmatrix} 1 & \cdot & 3 & 4 \\ 1 & 2 & \cdot & 8 \\ 3 & 2 & 3 & \cdot \\ 4 & 8 & 3 & 4 \\ \cdot & 10 & 15 & 4 \end{bmatrix}.$$

*We can verify that $\mathbf{X}^1$ and $\mathbf{X}^2$ are observed on uniquely completable sampling patterns for LRMC [3]. Nonetheless, we can construct the following* false *rank-1 matrices that agree with $\mathbf{X}_\Omega$:*

$$\tilde{\mathbf{X}}^1 = \begin{bmatrix} 60 & 40 & 15 & 4 \\ 1 & 2/3 & 1/4 & 1/15 \\ 3 & 2 & 3/4 & 1/5 \\ 12 & 8 & 3 & 4/5 \\ 60 & 40 & 15 & 4 \end{bmatrix}, \quad \tilde{\mathbf{X}}^2 = \begin{bmatrix} 1 & 1/4 & 3 & 1 \\ 8 & 2 & 24 & 8 \\ 1 & 1/4 & 3 & 1 \\ 4 & 1 & 12 & 4 \\ 40 & 10 & 120 & 40 \end{bmatrix}.$$

*This shows that even with unlimited computational power, if we exhaustively search all the identifiable patterns for LRMC, we can end up with false mixtures. Hence the importance of studying the identifiable patterns for MMC.*

False mixtures arise because we do not know a priori which entries of $\mathbf{X}_\Omega$ correspond to each $\mathbf{X}^k$. Hence, it is possible that a rank-r matrix $\tilde{\mathbf{X}}$ agrees with some entries from $\mathbf{X}^1$, other entries from $\mathbf{X}^2$, and so on. Furthermore, $\tilde{\mathbf{X}}$ may even be *the only* rank-r matrix that agrees with such combination of entries, as in Example 1.

**Remark 1.** *Recall that LRMC and HRMC are tantamount to identifying the subspace(s) containing the columns of $\mathbf{X}$ [3, 4]. In fact, if we knew such subspaces, LRMC and HRMC become almost trivial problems (see Appendix A for details). Similarly, if no data is missing, HRMC simplifies to subspace clustering, which has been studied extensively, and is now reasonably well-understood [53–62]. In contrast, MMC remains challenging even if the subspaces corresponding to the low-rank matrices in the mixture are known, and even $\mathbf{X}$ is fully observed. We refer the curious reader to Appendix A, and point out the bottom row and the last column in Figure 2, which show the MMC error when the underlying subspaces are known, and when $\mathbf{X}$ is fully observed.*

## 4  Main Theoretical Results

Example 1 shows the importance of studying the identifiable patterns for MMC, which we do now. First recall that $r + 1$ samples per column are necessary for LRMC [3]. This implies that even if an oracle told us $\boldsymbol{\Omega}^1, \dots, \boldsymbol{\Omega}^K$, if we intend to recover a column of $\mathbf{X}^k$, we need to observe it on at least $r + 1$ entries. Hence we assume without loss of generality that:

> **(A1)** Each column of $\boldsymbol{\Omega}^k$ has either $0$ or $r + 1$ non-zero entries.

In words, **A1** requires that each column of $\mathbf{X}^k$ to be recovered is observed on exactly $r + 1$ entries. Of course, observing more entries may only aid completion. Hence, rather than an assumption, **A1** describes the most difficult scenario where we have the bare minimum amount of information required for completion. We use **A1** to ease notation, exposition and analysis. All our results can be easily extended to the case where **A1** is dropped (see Remark 2).

Without further assumptions on $\mathbf{X}$, completion (of any kind) may be impossible. To see this consider the simple example where $\mathbf{X}$ is only supported on the $i^{\text{th}}$ row. Then it would be impossible to recover $\mathbf{X}$ unless all columns were observed on the $i^{\text{th}}$ row. In most completion applications this would be unlikely. For example, in a movies recommender system like Netflix, this would require that *all* the users watched (and rated) the same movie.

To rule out scenarios like these, typical completion theory requires incoherence and uniform sampling. Incoherence guarantees that the information is well-spread over the matrix. Uniform sampling guarantees that all rows and columns are sufficiently sampled. However, it is usually unclear (and generally unverifiable) whether an incomplete matrix is coherent. Furthermore, observations are hardly ever uniformly distributed. For instance, we do not expect children to watch adults movies.

To avoid these issues, instead of incoherence we will assume that $\mathbf{X}$ is a *generic* mixture of low-rank matrices. More precisely, we assume that:

> **(A2)** $\mathbf{X}^1, \dots, \mathbf{X}^K$ are drawn independently according to an absolutely continuous distribution with respect to the Lebesgue measure on the determinantal variety (set of all $d \times n$, rank-r matrices).

**A2** essentially requires that each $\mathbf{X}^k$ is a generic rank-r matrix. This type of *genericity* assumptions are becoming increasingly common in studies of LRMC, HRMC, and related problems [3, 4, 23–27, 46]. See Appendix C for a further discussion on **A2**, and its relation to other common assumptions from the literature.

With this, we are ready to present our main theorem. It gives a deterministic condition on $\boldsymbol{\Omega}$ to guarantee that $\mathbf{X}^1, \ldots, \mathbf{X}^K$ can be identified from $\mathbf{X}_{\boldsymbol{\Omega}}$. This provides information-theoretic requirements for MMC. The proof is in Appendix B.

---

**Theorem 1.** *Let* **A1**-**A2** *hold. Suppose there exist matrices* $\{\boldsymbol{\Omega}_\tau\}_{\tau=1}^{r+1}$ *formed with disjoint subsets of* $(d - r + 1)$ *columns of* $\boldsymbol{\Omega}^k$*, such that for every* $\tau$*:*

(†) *Every matrix* $\boldsymbol{\Omega}'$ *formed with a* proper *subset of the columns in* $\boldsymbol{\Omega}_\tau$ *has at least* r *fewer columns than non-zero rows.*

*Then all the columns of* $\mathbf{X}^k$ *that have observations in* $\mathbf{X}_{\boldsymbol{\Omega}}$ *are identifiable.*

---

In words, Theorem 1 states that MMC is possible as long as we observe *the right entries* in each $\mathbf{X}^k$. The intuition is that each of these entries imposes a constraint on what $\mathbf{X}^1, \ldots, \mathbf{X}^K$ may be, and the pattern in $\boldsymbol{\Omega}$ determines whether these constraints are redundant. Patterns satisfying the conditions of Theorem 1 guarantee that $\mathbf{X}^1, \ldots, \mathbf{X}^K$ is the only mixture that satisfies the constraints produced by the observed entries.

**Remark 2.** *Recall that* r + 1 *samples per column are strictly necessary for completion.* **A1** *requires that we have exactly that minimum number of samples. If* $\mathbf{X}^k$ *is observed on more than* r + 1 *entries per column, it suffices that* $\boldsymbol{\Omega}^k$ *contains a pattern satisfying the conditions of Theorem 1.*

Theorem 1 shows that MMC is possible if the samplings satisfy certain combinatorial conditions. Our next result shows that if each entry of $\mathbf{X}^k$ is observed on $\mathbf{X}_{\boldsymbol{\Omega}}$ with probability $\mathcal{O}(\frac{1}{d}\max\{r, \log d\})$, then with high probability $\boldsymbol{\Omega}^k$ will satisfy such conditions. The proof is in Appendix B.

---

**Theorem 2.** *Suppose* $r \leq \frac{d}{6}$ *and* $n \geq (r+1)(d-r+1)$*. Let* $\epsilon > 0$ *be given. Suppose that an entry of* $\mathbf{X}_{\boldsymbol{\Omega}}$ *is equal to the corresponding entry of* $\mathbf{X}^k$ *with probability*

$$p \geq \frac{2}{d}\max\left\{2r,\ 12\left(\log(\frac{d}{\epsilon})+1\right)\right\}.$$

*Then* $\boldsymbol{\Omega}^k$ *satisfies the sampling conditions of Theorem 1 with probability* $\geq 1 - 2(r+1)\epsilon$*.*

---

Theorem 2 shows that the sample complexity of MMC is $\mathcal{O}(K\max\{r, \log d\})$ observations per column of $\mathbf{X}_{\boldsymbol{\Omega}}$. This is exactly the same as the sample complexity of HRMC [4], and simplifies to $\mathcal{O}(\max\{r, \log d\})$ if $K = 1$, corresponding to the sample complexity of LRMC [3]. Intuitively, this means that information-theoretically, we virtually pay no price for mixing low-rank matrices.

## 5 Alternating Algorithm for MMC

Theorems 1 and 2 show that MMC is theoretically possible under reasonable conditions (virtually the same as LRMC and HRMC). However, these results follow from a combinatorial analysis that is infeasible in practice (see Appendix B for details). To address this, we derive a practical alternating algorithm for MMC, which we call AMMC (alternating mixture matrix completion).

The main idea is that MMC, like most mixture problems, can be viewed as a clustering task: if we could determine the entries of $\mathbf{X}_{\boldsymbol{\Omega}}$ that correspond to each $\mathbf{X}^k$, then we would be able to partition $\mathbf{X}_{\boldsymbol{\Omega}}$ into K incomplete low-rank matrices, and then complete them using standard LRMC. The question is how to determine which entries of $\mathbf{X}_{\boldsymbol{\Omega}}$ correspond to each $\mathbf{X}^k$, i.e., how to determine $\boldsymbol{\Omega}^1, \ldots, \boldsymbol{\Omega}^K$. To address this, let $\mathbf{U}^k \in \mathbb{R}^{d \times r}$ be a basis for the subspace containing the columns of $\mathbf{X}^k$, and let $\mathbf{x}_{\boldsymbol{\omega}}$ denote the $j^{\text{th}}$ column of $\mathbf{X}_{\boldsymbol{\Omega}}$, observed only on the entries indexed by $\boldsymbol{\omega} \subset \{1, \ldots, d\}$. For any subspace, matrix or vector that is compatible with a set of indices $\cdot$, we use the subscript $\cdot$ to denote

its restriction to the coordinates/rows in $\cdot$. For example, $\mathbf{U}_{\boldsymbol{\omega}}^k \in \mathbb{R}^{|\boldsymbol{\omega}| \times r}$ denotes the restriction of $\mathbf{U}^k$ to the indices in $\boldsymbol{\omega}$. Suppose $\mathbf{x}_{\boldsymbol{\omega}}$ contains entries from $\mathbf{X}^k$, and let $\boldsymbol{\omega}^k \subset \boldsymbol{\omega}$ index such entries. Then our goal is to determine $\boldsymbol{\omega}^k$, as that would tell us the $j^{\text{th}}$ column of $\boldsymbol{\Omega}^k$. Since $\mathbf{x}_{\boldsymbol{\omega}^k} \in \text{span}\{\mathbf{U}_{\boldsymbol{\omega}^k}^k\}$, we can restate our goal as finding the set $\boldsymbol{\omega}^k \subset \boldsymbol{\omega}$ such that $\mathbf{x}_{\boldsymbol{\omega}^k} \in \text{span}\{\mathbf{U}_{\boldsymbol{\omega}^k}^k\}$.

To find $\boldsymbol{\omega}^k$, let $\boldsymbol{v} \subset \boldsymbol{\omega}$, and let $\mathbf{P}_{\boldsymbol{v}}^k := \mathbf{U}_{\boldsymbol{v}}^k (\mathbf{U}_{\boldsymbol{v}}^{k\mathsf{T}} \mathbf{U}_{\boldsymbol{v}}^k)^{-1} \mathbf{U}_{\boldsymbol{v}}^{k\mathsf{T}}$ denote the projection operator onto $\text{span}\{\mathbf{U}_{\boldsymbol{v}}^k\}$. Recall that $\|\mathbf{P}_{\boldsymbol{v}}^k \mathbf{x}_{\boldsymbol{v}}\| \leq \|\mathbf{x}_{\boldsymbol{v}}\|$, with equality if and only if $\mathbf{x}_{\boldsymbol{v}} \in \text{span}\{\mathbf{U}_{\boldsymbol{v}}^k\}$. It follows that $\boldsymbol{\omega}^k$ is the largest set $\boldsymbol{v}$ such that $\|\mathbf{P}_{\boldsymbol{v}}^k \mathbf{x}_{\boldsymbol{v}}\| = \|\mathbf{x}_{\boldsymbol{v}}\|$. In other words, $\boldsymbol{\omega}^k$ is the solution to

$$\underset{\boldsymbol{v} \subset \boldsymbol{\omega}}{\arg\max} \quad \|\mathbf{P}_{\boldsymbol{v}}^k \mathbf{x}_{\boldsymbol{v}}\| - \|\mathbf{x}_{\boldsymbol{v}}\| + |\boldsymbol{v}|. \tag{1}$$

However, (1) is non-convex. Hence, in order to find the solution to (1), we propose the following *erasure* strategy. The main idea is to start our search with $\boldsymbol{v} = \boldsymbol{\omega}$, and then iteratively remove the entries (coordinates) of $\boldsymbol{v}$ that most increase the gap between $\|\mathbf{P}_{\boldsymbol{v}}^k \mathbf{x}_{\boldsymbol{v}}\|$ and $\|\mathbf{x}_{\boldsymbol{v}}\|$ (hence the term *erasure*). We stop this procedure when $\|\mathbf{P}_{\boldsymbol{v}}^k \mathbf{x}_{\boldsymbol{v}}\|$ is equal to $\|\mathbf{x}_{\boldsymbol{v}}\|$ (or close enough). More precisely, we initialize $\boldsymbol{v} = \boldsymbol{\omega}$, and then iteratively redefine $\boldsymbol{v}$ as the set

$$\boldsymbol{v} = \boldsymbol{v} \backslash \mathrm{i}, \quad \text{where} \quad \mathrm{i} = \underset{i \in \boldsymbol{v}}{\arg\max} \quad \|\mathbf{P}_{\boldsymbol{v} \backslash i}^k \mathbf{x}_{\boldsymbol{v} \backslash i}\| - \|\mathbf{x}_{\boldsymbol{v} \backslash i}\|. \tag{2}$$

In words, $\mathrm{i}$ is the coordinate of the vector $\mathbf{x}_{\boldsymbol{v}}$ such that if ignored, the gap between the remaining vector $\mathbf{x}_{\boldsymbol{v} \backslash \mathrm{i}}$ and its projection $\mathbf{P}_{\boldsymbol{v} \backslash \mathrm{i}}^k \mathbf{x}_{\boldsymbol{v} \backslash \mathrm{i}}$ is reduced the most. At each iteration we remove (erase) such coordinate $\mathrm{i}$ from $\boldsymbol{v}$. The intuition behind this approach is that the coordinates of $\mathbf{x}_{\boldsymbol{v}}$ that do not correspond to $\mathbf{X}^k$ are more likely to increase the gap between $\|\mathbf{P}_{\boldsymbol{v}}^k \mathbf{x}_{\boldsymbol{v}}\|$ and $\|\mathbf{x}_{\boldsymbol{v}}\|$. Notice that if $\mathbf{U}^k$ is in general position (guaranteed by **A2**) and $|\boldsymbol{v}| \leq r$, then $\mathbf{U}_{\boldsymbol{v}}^k = \mathbb{R}^{|\boldsymbol{v}|}$ (because $\mathbf{U}^k$ is r-dimensional). In such case, it is trivially true that $\mathbf{x}_{\boldsymbol{v}} \in \text{span}\{\mathbf{U}_{\boldsymbol{v}}^k\}$, whence $\|\mathbf{P}_{\boldsymbol{v}}^k \mathbf{x}_{\boldsymbol{v}}\| = \|\mathbf{x}_{\boldsymbol{v}}\|$. Hence the procedure above is guaranteed to terminate after at most $\|\boldsymbol{\omega}\| - r$ iterations. At such point, $|\boldsymbol{v}| = r$, and we know that we were unable to find $\boldsymbol{\omega}^k$ (or a subset of it). One alternative is to start with a different $\boldsymbol{v}_0 \subsetneq \boldsymbol{\omega}$, and search again.

This procedure may remove some entries from $\boldsymbol{\omega}^k$ along the way, so in general, the output of this process will be a set $\boldsymbol{v} \subset \boldsymbol{\omega}^k$. However, finding a subset of $\boldsymbol{\omega}^k$ is enough to find $\boldsymbol{\omega}^k$. To see this, recall that since $\mathbf{x}_{\boldsymbol{\omega}^k} \in \text{span}\{\mathbf{U}_{\boldsymbol{\omega}^k}^k\}$, there is a coefficient vector $\boldsymbol{\theta}^k \in \mathbb{R}^r$ such that $\mathbf{x}_{\boldsymbol{\omega}^k} = \mathbf{U}_{\boldsymbol{\omega}^k}^k \boldsymbol{\theta}^k$. Since $\boldsymbol{v} \subset \boldsymbol{\omega}^k$, it follows that $\mathbf{x}_{\boldsymbol{v}} = \mathbf{U}_{\boldsymbol{v}}^k \boldsymbol{\theta}^k$. Furthermore, since $|\boldsymbol{v}| \geq r$, we can find $\boldsymbol{\theta}^k$ as $\boldsymbol{\theta}^k = (\mathbf{U}_{\boldsymbol{v}}^{k\mathsf{T}} \mathbf{U}_{\boldsymbol{v}}^k)^{-1} \mathbf{U}_{\boldsymbol{v}}^{k\mathsf{T}} \mathbf{x}_{\boldsymbol{v}}$. Since $\mathbf{x}_{\boldsymbol{\omega}^k} = \mathbf{U}_{\boldsymbol{\omega}^k}^k \boldsymbol{\theta}^k$, at this point we can identify $\boldsymbol{\omega}^k$ by simple inspection (the matching entries in $\mathbf{x}_{\boldsymbol{\omega}}$ and $\mathbf{U}_{\boldsymbol{\omega}}^k \boldsymbol{\theta}^k$). Recall that $\boldsymbol{\omega}^k$ determines the $j^{\text{th}}$ column of $\boldsymbol{\Omega}^k$. Hence, if we repeat the procedure above for each column in $\mathbf{X}_{\boldsymbol{\Omega}}$ and each k, we can recover $\boldsymbol{\Omega}^1, \ldots, \boldsymbol{\Omega}^K$. After this, we can use standard LRMC on $\mathbf{X}_{\boldsymbol{\Omega}^1}, \ldots, \mathbf{X}_{\boldsymbol{\Omega}^K}$ to recover $\mathbf{X}^1, \ldots \mathbf{X}^K$ (which is the ultimate goal of MMC).

The catch here is that this procedure requires knowing $\mathbf{U}^k$, which we do not know. So essentially we have a chicken and egg problem: (i) if we knew $\mathbf{U}^k$, we would be able to find $\boldsymbol{\Omega}^k$. (ii) If we knew $\boldsymbol{\Omega}^k$ we would be able to find $\mathbf{U}^k$ (and $\mathbf{X}^k$, using standard LRMC on $\mathbf{X}_{\boldsymbol{\Omega}^k}$). Since we know neither, we use a common technique for these kind of problems: alternate between finding $\boldsymbol{\Omega}^k$ and $\mathbf{U}^k$. More precisely, we start with some initial guesses $\hat{\mathbf{U}}^1, \ldots, \hat{\mathbf{U}}^K$, and then alternate between the following two steps until convergence:

(i) **Cluster.** Let $\mathbf{x}_{\boldsymbol{\omega}}$ be the $j^{\text{th}}$ column in $\mathbf{X}_{\boldsymbol{\Omega}}$. For each $k = 1, \ldots, K$, we first *erase* entries from $\boldsymbol{\omega}$ to obtain a set $\boldsymbol{v} \subset \boldsymbol{\omega}$ indicating entries likely to correspond to $\mathbf{X}^k$. This *erasure* procedure initializes $\boldsymbol{v} = \boldsymbol{\omega}$, and then repeats (2), (replacing $\mathbf{P}^k$ with $\hat{\mathbf{P}}^k$, which denotes the projection operator onto $\text{span}\{\hat{\mathbf{U}}^k\}$) until we to obtain a set $\boldsymbol{v} \subset \boldsymbol{\omega}$ such that the projection $\|\hat{\mathbf{P}}_{\boldsymbol{v}}^k \mathbf{x}_{\boldsymbol{v}}\|$ is close to $\|\mathbf{x}_{\boldsymbol{v}}\|$. This way, the entries of $\mathbf{x}_{\boldsymbol{v}}$ are likely to correspond to $\mathbf{X}^k$. Using these entries, we can estimate the coefficient of the $j^{\text{th}}$ column of $\mathbf{X}^k$ with respect to $\mathbf{U}^k$, given by $\hat{\boldsymbol{\theta}}^k = (\hat{\mathbf{U}}_{\boldsymbol{v}^k}^{k\mathsf{T}} \hat{\mathbf{U}}_{\boldsymbol{v}^k}^k)^{-1} \hat{\mathbf{U}}_{\boldsymbol{v}^k}^{k\mathsf{T}} \mathbf{x}_{\boldsymbol{v}^k}$. With $\hat{\boldsymbol{\theta}}^k$ we can also estimate the $j^{\text{th}}$ column of $\mathbf{X}^k$ as $\hat{\mathbf{x}}^k := \hat{\mathbf{U}}^k \hat{\boldsymbol{\theta}}^k$. Notice that both $\boldsymbol{v}$ and $\hat{\mathbf{x}}^k$ are obtained using $\hat{\mathbf{U}}^k$, which may be different from $\mathbf{U}^k$. It follows that $\boldsymbol{v}$ may contain some entries that do not correspond to $\mathbf{X}^k$, and $\hat{\mathbf{x}}^k$ may be inaccurate. Hence, in general, $\mathbf{x}_{\boldsymbol{\omega}}$ and $\hat{\mathbf{x}}_{\boldsymbol{\omega}}^k$ will have no matching entries, and so we cannot identify $\boldsymbol{\omega}^k$ by simple inspection, as before. However, we can repeat our procedure for each k to obtain estimates $\hat{\mathbf{x}}_{\boldsymbol{\omega}}^1, \ldots, \hat{\mathbf{x}}_{\boldsymbol{\omega}}^K$, and then assign each entry of $\mathbf{x}_{\boldsymbol{\omega}}$ to its closest match. More

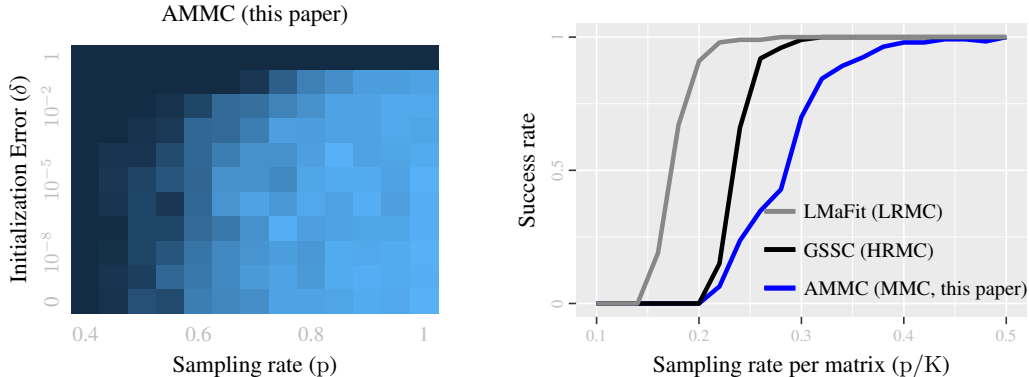

Figure 2: **Left:** Success rate (average over 100 trials) of AMMC as a function of the fraction of observed entries p and the distance $\delta$ between the *true* subspaces $\mathbf{U}^k$ and their initial estimates. Lightest represents 100% success rate; darkest represents 0%. **Right:** Comparison of state-of-the-art algorithms for LRMC, HRMC, and MMC (in their respective settings; see Figure 1). The performance of AMMC (in the more difficult problem of MMC) is comparable to the performance of state-of-the-art algorithms in the simpler problems of LRMC and HRMC.

precisely, our estimate $\hat{\boldsymbol{\omega}}^k \subset \boldsymbol{\omega}$ (indicating the entries of $\mathbf{x}_{\boldsymbol{\omega}}$ that we estimate that correspond to $\mathbf{X}^k$) will contain entry $i \in \boldsymbol{\omega}$ if $|x_i - \hat{x}_i^k| \leq |x_i - \hat{x}_i^\ell|$ for every $\ell = 1, \dots, K$. Repeating this procedure for each column of $\mathbf{X}_{\boldsymbol{\Omega}}$ will produce estimates $\hat{\boldsymbol{\Omega}}^1, \dots, \hat{\boldsymbol{\Omega}}^K$. Specifically, the $j^{th}$ column of $\hat{\boldsymbol{\Omega}}^k \in \{0,1\}^{d \times n}$ will contain a 1 in the rows indicated by $\hat{\boldsymbol{\omega}}^k$.

(ii) **Complete.** For each $k$, complete $\mathbf{X}_{\hat{\boldsymbol{\Omega}}^k}$ using your favorite LRMC algorithm. Then compute a new estimate $\hat{\mathbf{U}}^k$ given by the leading $r$ left singular vectors of the completion of $\mathbf{X}_{\hat{\boldsymbol{\Omega}}^k}$.

The entire procedure is summarized in Algorithm 1, in Appendix D, where we also discuss initialization, generalizations to noise and outliers, and other simple extensions to improve performance.

## 6 Experiments

**Simulations.** We first present a series of synthetic experiments to study the performance of AMMC (Algorithm 1). In our simulations we first generate matrices $\mathbf{U}^k \in \mathbb{R}^{d \times r}$ and $\boldsymbol{\Theta}^k \in \mathbb{R}^{r \times n}$ with i.i.d. $\mathcal{N}(0,1)$ entries to use as bases and coefficients of the low-rank matrices in the mixture, i.e., $\mathbf{X}^k = \mathbf{U}^k \boldsymbol{\Theta}^k \in \mathbb{R}^{d \times n}$. Here $d = n = 100$, $r = 5$ and $K = 2$. With probability $(1-p)$, the $(i,j)^{th}$ entry of $\mathbf{X}_{\boldsymbol{\Omega}}$ will be missing, and with probability $p/K$ it will be equal to the corresponding entry in $\mathbf{X}^k$. Recall that similar to EM and other alternating approaches, AMMC depends on initialization. Hence, we study the performance of AMMC as a function of both $p$ and the distance $\delta \in [0,1]$ between $\{\mathbf{U}^k\}$ and their initial estimates (measured as the normalized Frobenius norm of the difference between their projection operators). We measure accuracy using the normalized Frobenius norm of the difference between each $\mathbf{X}^k$ and its completion. We considered a success if this quantity was below $10^{-8}$. The results of 100 trials are summarized in Figure 2.

Notice that the performance of AMMC decays nicely with the distance $\delta$ between the *true* subspaces $\mathbf{U}^k$ and their initial estimates. We can see this type of behavior in similar state-of-the-art alternating algorithms for the simpler problem of HRMC [19]. Since MMC is highly non-convex, it is not surprising that if the initial estimates are poor (far from the truth), then AMMC may converge to a local minimum. Similarly, the performance of AMMC decays nicely with the fraction of observed entries $p$. Notice that even if $\mathbf{X}$ is fully observed ($p = 1$), if the initial estimates are very far from the true subspaces ($\delta = 1$), then AMMC performs poorly. This shows, consistent with our discussing in Remark 1, that in practice MMC is a challenging problem even if $\mathbf{X}$ is fully observed. Hence, it is quite remarkable that AMMC works most of the time with as little as $p \approx 0.6$, corresponding to observing $\approx 0.3$ of the entries in each $\mathbf{X}^k$. To put this under perspective, notice (Figure 2) that this is comparable the amount of missing data tolerated by GSSC [19] and LMaFit [11], which are state-of-the-art for the simpler problems of HRMC (special case of MMC where all entries in each column of $\mathbf{X}$ correspond to the same $\mathbf{X}^k$) and LRMC (special case where there is only one $\mathbf{X}^k$).

| Mixture | —— Reconstructions —— | Original | Robust PCA | MMC (this paper) |

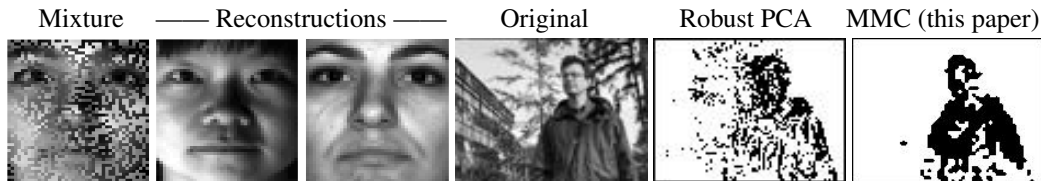

Figure 3: **Left 3:** Reconstructed images from a mixture. **Right 3:** Original frame and segmented foreground.

To obtain Figure 2 we replicated the same setup as above, but with data generated according to the HRMC and LRMC models. Hence, we conclude that the performance of AMMC (in the more difficult problem of MMC) is comparable to the performance of state-of-the-art algorithms for the much simpler problems of HRMC and LRMC.

We point out that according to Theorems 1 and 2, MMC is theoretically possible with $p \geq 1/2$. However, we can see that (even if $\mathbf{U}^1, \ldots, \mathbf{U}^K$ are known, corresponding to $\delta = 0$ in Figure 2) the performance of AMMC is quite poor if $p < 0.6$. This shows two things: (i) MMC is challenging even if $\mathbf{U}^1, \ldots, \mathbf{U}^K$ are known (as discussed in Remark 1), and (ii) there is a gap between what is information-theoretically possible and what is currently possible in practice (with AMMC). In future work we will explore algorithms that can approach the information-theoretic limits.

**Real Data: Face Clustering and Inpainting.** It is well-known that images of an individual's face are approximately low-rank [63]. Natural images, however, usually contain faces of multiple individuals, often partially occluding each other, resulting in a mixture of low-rank matrices. In this experiment we demonstrate the power of MMC in two tasks: first, classifying partially occluded faces in an image, and second, image inpainting [47]. To this end, we use the Yale B dataset [64], containing 2432 photos of 38 subjects (64 photos per subject), each photo of size $48 \times 42$. We randomly select two subjects, and vectorize and concatenate their images to obtain two approximately rank-10 matrices $\mathbf{X}^1, \mathbf{X}^2 \in \mathbb{R}^{2016 \times 64}$. Next we combine them into $\mathbf{X} \in \mathbb{R}^{2016 \times 64}$, whose each entry is equal to the corresponding entry in $\mathbf{X}^1$ or $\mathbf{X}^2$ with equal probability. This way, each column of $\mathbf{X}$ contains a mixed image with pixels from multiple individuals. We aim at two goals: (i) classify the entries in $\mathbf{X}$ according to $\mathbf{X}^1$ and $\mathbf{X}^2$, which in turn means locating and classifying the face of each individual in each image, and (ii) recover $\mathbf{X}^1$ and $\mathbf{X}^2$ from $\mathbf{X}$, thus reconstructing the unobserved pixels in each image (inpainting). We repeat this experiment 30 times using AMMC (with gaussian random initialization, known to produce near-orthogonal subspaces with high probability), obtaining a pixel classification error of $2.98\%$, and a reconstruction error of $4.1\%$, which is remarkable in light that the ideal rank-10 approximation (no mixture, and full data) achieves $1.8\%$. Figure 3 shows an example, with more in Figure 4 in Appendix E. Notice that in this case we cannot compare against other methods, as AMMC is the first, and currently the only method for MMC.

**Real Data: MMC for Background Segmentation.** As discussed in Section 2, Robust PCA models a video as the superposition of a low-rank background plus a sparse foreground with no structure. MMC brings more flexibility, allowing multiple low-rank matrices to model background, structured foreground objects (sparse or abundant) and illumination artifacts, while at the same time also accounting for outliers (the entries/pixels that were assigned to no matrix in the mixture). In fact, contrary to Robust PCA, MMC allows a very large (even dominant) fraction of outliers. In this experiment we test AMMC in the task of background segmentation, using the Wallflower [65] and the I2R [66] datasets, containing videos of traffic cameras, lobbies, and pedestrians in the street. For each video, we compare AMMC (with gaussian random initialization) against the best result amongst the following state-of-the-art algorithms for Robust PCA: [35–39]. We chose these methods based on the comprehensive review in [40], and previous reports [41–43] indicating that these algorithms typically performed as well or better than several others, including [44, 45]. In most cases, both Robust PCA and AMMC perform quite similarly (see Figure 5 in Appendix E). However, in one case AMMC achieves $87.67\%$ segmentation accuracy (compared with the ground truth, manually segmented), while Robust PCA only achieves $74.88\%$ (Figure 3). Our hypothesis is that this is due to the large portion of outliers (foreground). It is out of the scope of this paper, but of interest for future work, to collect real datasets with similar properties, where AMMC can be further tested. We point out, however, that AMMC is orders of magnitude slower than Robust PCA. Our future work will also focus on developing faster methods for MMC.

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
