[Supplementary Material · MixtureMC_supplement.pdf]

# A MMC with Known Subspaces and Full Data

Remark 1 points out that if we knew the subspace(s) containing the columns in $\mathbf{X}$, then LRMC and HRMC become almost trivial problems, while MMC still remains challenging. To see this, recall that by definition, the columns in a rank-r matrix lie in an r-dimensional subspace. Recall that $\mathbf{x}_\omega$ denotes the $\mathrm{j}^{\mathrm{th}}$ column of $\mathbf{X}_\Omega$, observed only on the entries indexed by $\omega \subset \{1, \ldots, \mathrm{d}\}$, and that $\mathbf{U}^{\mathrm{k}} \in \mathbb{R}^{\mathrm{d} \times \mathrm{r}}$ spans the subspace containing the columns of $\mathbf{X}^{\mathrm{k}}$.

Next suppose that all the entries in $\mathbf{x}_\omega$ correspond to the same subspace (as would be the case in HRMC and LRMC), and that we know $\mathbf{U}^1, \ldots, \mathbf{U}^{\mathrm{K}}$. Then we can project $\mathbf{x}_\omega$ onto the subspaces generated by $\mathbf{U}^1_\omega, \ldots, \mathbf{U}^{\mathrm{K}}_\omega$ to determine which subspace $\mathbf{x}_\omega$ corresponds to. Say it is $\mathbf{U}^{\mathrm{k}}$. Then we can compute the coefficient of $\mathbf{x}_\omega$ as $\boldsymbol{\theta} = (\mathbf{U}^{\mathrm{kT}}_\omega \mathbf{U}^{\mathrm{k}}_\omega)^{-1} \mathbf{U}^{\mathrm{kT}}_\omega \mathbf{x}_\omega$. Since the coefficient of $\mathbf{x}$ is the same as the coefficient of $\mathbf{x}_\omega$, we can recover $\mathbf{x} = \mathbf{U}^{\mathrm{k}} \boldsymbol{\theta}$.

In contrast, in MMC the entries in $\mathbf{x}_\omega$ may belong to multiple subspaces, and hence, even if we know $\mathbf{U}^1, \ldots, \mathbf{U}^{\mathrm{K}}$, we cannot just project to identify the subspace corresponding to $\mathbf{x}_\omega$ (if $\mathbf{x}_\omega$ has entries from more than one subspace, it will not lie in *any* of the K subspaces). Hence, MMC can be very challenging even if we know $\mathbf{U}^1, \ldots, \mathbf{U}^{\mathrm{K}}$. This can be seen in our experiments. In particular pay attention to the bottom row in Figure 2, which shows the MMC error when $\mathbf{U}^1, \ldots, \mathbf{U}^{\mathrm{K}}$ are known.

Similarly, MMC is difficult even if $\mathbf{X}$ is fully observed! To build some intuition, consider HRMC. If no data is missing, HRMC simplifies to *subspace clustering* (SC) [53], which has been studied extensively in recent years to produce theory and algorithms to handle gross errors [54–58], noise [59], privacy [60] and data constraints [61]. Furthermore, the renowned state-of-the-art algorithm *sparse subspace clustering* [62], can efficiently, accurately and provably perform SC. Hence, if $\mathbf{X}$ is fully observed, HRMC is well understood.

In contrast, even if $\mathbf{X}$ is fully observed, MMC remains MMC, because we still do not know which entries belong together, and because for each entry in $\mathbf{X}$ that we observe, there are $\mathrm{K} - 1$ that we do not. For example, if we observe an entry of $\mathbf{X}$ corresponding to $\mathbf{X}^1$, we still do not know that it belongs to $\mathbf{X}^1$, and we still need to recover the corresponding entries of $\mathbf{X}^2, \ldots, \mathbf{X}^{\mathrm{K}}$. Furthermore, as we discussed above, and in Section 5, even if $\mathbf{U}^1, \ldots, \mathbf{U}^{\mathrm{K}}$ were known, identifying the entries that agree with the subspace is not a trivial problem. Hence, MMC remains a challenging problem even with *full* data. This can be seen in our experiments. In particular pay attention to the last column in Figure 2, which shows the MMC error when $\mathbf{X}$ is fully observed.

# B Proofs

As discussed in Section 3, the main subtlety in MMC is that since we do not know a priori which entries of $\mathbf{X}_\Omega$ correspond to each $\mathbf{X}^{\mathrm{k}}$, there could arise *false* mixtures that agree with $\mathbf{X}_\Omega$. Fortunately, Theorem 3 in [4] gives conditions to guarantee that a subset of entries correspond to the same $\mathbf{X}^{\mathrm{k}}$. We restate this result as the following Lemma, with some adaptations to our context.

**Lemma 1** (Theorem 3 in [4]). *Let **A2** hold. Let $\mathbf{X}', \mathbf{X}^\tau$ be matrices formed with disjoint subsets of the columns in $\mathbf{X}$. Let $\mathbf{\Omega}', \mathbf{\Omega}_\tau$ indicate subsets of the observed entries in $\mathbf{X}'$ and $\mathbf{X}^\tau$ with at least $\mathrm{r} + 1$ samples per column. Suppose there are only finitely many rank-r matrices that agree with $\mathbf{X}'_{\mathbf{\Omega}'}$, and that $\mathbf{\Omega}_\tau \in \{0, 1\}^{\mathrm{d} \times (\mathrm{d-r+1})}$ satisfies condition (†) in Theorem 1. If there is a rank-r matrix that agrees with $[\mathbf{X}'_{\mathbf{\Omega}'} \ \mathbf{X}^\tau_{\mathbf{\Omega}_\tau}]$, then such matrix is unique, and all entries in $[\mathbf{X}'_{\mathbf{\Omega}'} \ \mathbf{X}^\tau_{\mathbf{\Omega}_\tau}]$ correspond to the same $\mathbf{X}^{\mathrm{k}}$.*

The main insight behind Lemma 1 is that the observed entries in $\mathbf{X}'_{\mathbf{\Omega}'}$ impose restrictions on the rank-r matrices that may agree with the observations. The restrictions produced by $\mathbf{X}'_{\mathbf{\Omega}'}$ may be enough to narrow the possible solutions to a finite number of options. However, some of these restrictions may come from $\mathbf{X}^1$, others from $\mathbf{X}^2$, and so on. In such case, it is possible that the combined restrictions are compatible, leading to false rank-r matrices that agree with $\mathbf{X}'_{\mathbf{\Omega}'}$. Incorporating $\mathbf{X}^\tau_{\mathbf{\Omega}_\tau}$ adds more restrictions. The sampling pattern in $\mathbf{\Omega}_\tau$ guarantees that the new restrictions will add enough redundancy, such that if the restrictions do not come from the same $\mathbf{X}^{\mathrm{k}}$, they will be inconsistent, implying that no rank-r matrix can possibly agree with $[\mathbf{X}'_{\mathbf{\Omega}'} \ \mathbf{X}^\tau_{\mathbf{\Omega}_\tau}]$. Intuitively, $\mathbf{X}^\tau_{\mathbf{\Omega}_\tau}$ works as a *checksum* matrix.

Lemma 1 requires that $\mathbf{X}'_{\Omega'}$ is finitely completable. Theorem 1 and Lemma 1 in [3] give conditions on $\Omega'$ to guarantee that this is the case. We combine these results in the following Lemma, with some adaptations to our context.

**Lemma 2** (Theorem 1 and Lemma 1 in [3]). *Let* **A2** *hold. Suppose* $\Omega'$ *can be partitioned into* $r$ *matrices* $\{\Omega_\tau\}_{\tau=1}^{r}$, *each of size* $d \times (d - r + 1)$, *such that condition* (†) *in Theorem 1 holds for every* $\tau$. *Then there are at most finitely many rank-$r$ matrices that agree with* $\mathbf{X}'_{\Omega'}$.

To summarize: Lemma 2 gives us conditions to guarantee that there are only finitely many rank-$r$ matrices that agree with a subset of entries. If these conditions are met, Lemma 1 provides further conditions to guarantee that there is only one such rank-$r$ matrix, and that all observations come from the same $\mathbf{X}^k$. Theorem 1 simply requires that each $\Omega^k$ satisfies the conditions of Lemmas 1 and 2. This way, we can just exhaustively search for all combinations of samplings that satisfy these conditions, knowing by assumption that we will eventually find $\Omega^1, \ldots, \Omega^K$. Then Lemmas 1 and 2 guarantee that we will be able to recover $\mathbf{X}^1, \ldots, \mathbf{X}^K$, and that we will find nothing else, i.e., no false mixtures.

***Proof of Theorem 1***. We will exhaustively search all combinations of samplings $\tilde{\Omega}$ with $(r+1)(d-r+1)$ columns of $\Omega$ and $r+1$ non-zero entries per column. For each such $\tilde{\Omega}$ we will verify whether it can be partitioned into matrices $\{\Omega_\tau\}_{\tau=1}^{r+1}$ satisfying (†). If so, we will verify whether there is a rank-$r$ matrix that agrees with $\tilde{\mathbf{X}}_{\tilde{\Omega}}$. In this case, Lemma 2 implies that $\tilde{\mathbf{X}}_{\tilde{\Omega}}$ is finitely completable (because $\{\Omega_\tau\}_{\tau=1}^{r}$ satisfy (†)). Furthermore, since $\Omega^{r+1}$ also satisfies (†), Lemma 1 implies that $\tilde{\mathbf{X}}_{\tilde{\Omega}}$ is uniquely completable, and that all its entries correspond to the same $\mathbf{X}^k$. It follows that $\mathbf{X}^k$ is the only rank-$r$ matrix that agrees with $\tilde{\mathbf{X}}_{\tilde{\Omega}}$.

By assumption, each $\Omega^k$ can be partitioned into matrices $\{\Omega_\tau\}_{\tau=1}^{r+1}$ satisfying (†). Hence the output of the procedure above will partition $\mathbf{X}_\Omega$ into $\mathbf{X}_{\Omega^1}, \ldots, \mathbf{X}_{\Omega^K}$. By **A1** each column in $\mathbf{X}_{\Omega^k}$ has either 0 or $r+1$ observations, so by Lemmas 1 and 2 we can recover all columns of $\mathbf{X}^k$ that have observations in $\mathbf{X}_\Omega$ using LRMC techniques [3]. $\square$

We now proceed to prove Theorem 2, which states that if an entry of $\mathbf{X}^k$ is observed with probability $p = \mathcal{O}(\frac{1}{d} \max\{r, \log d\})$, then with high probability $\Omega^k$ will satisfy the combinatorial conditions of Theorem 1, guaranteeing that $\mathbf{X}^k$ is identifiable. To this end, we will use the following lemma, stating that if $\mathbf{X}^k$ is observed on enough entries per column, then it will satisfy the combinatorial conditions of Theorem 2.

**Lemma 3.** *Suppose* $r \leq \frac{d}{6}$. *Let* $\epsilon > 0$ *be given. Suppose that* $\mathbf{X}^k$ *has at least* $(r+1)(d-r+1)$ *columns, each observed on at least* $m$ *locations, distributed uniformly at random, and independently across columns, with*

$$m \geq \max\left\{2r, \ 12\left(\log(\tfrac{d}{\epsilon})+1\right)\right\}. \tag{3}$$

*Then with probability at least* $1 - (r+1)\epsilon$, $\Omega^k$ *satisfies the sampling conditions of Theorem 1.*

Fortunately, we can prove Lemma 3 using Lemma 9 in [3], which we restate here with some adaptations as follows.

**Lemma 4.** *[Lemma 9 in [3]] Let the sampling assumptions of Lemma 3 hold. Let* $\Omega_{\tau-j}$ *be a matrix formed with* $d-r$ *columns of* $\Omega^k$. *Then with probability at least* $1 - \frac{\epsilon}{d}$, *every matrix* $\Omega'$ *formed with a subset of the columns in* $\Omega_{\tau-j}$ *(including* $\Omega_{\tau-j}$*) has at least* $r$ *fewer columns than non-zero rows.*

With Lemma 4, the proof of Lemma 3 follows by two union bounds.

***Proof of Lemma 3***. Randomly select $r+1$ disjoint matrices $\{\Omega_\tau\}_{\tau=1}^{r+1}$ from $\Omega^k$, each with $d-r+1$ columns. Let $\Omega_{\tau-j}$ denote the matrix formed with all but the $j^{\text{th}}$ column of $\Omega_\tau$. Using a union bound and Lemma 4, we can bound the probability that $\Omega_\tau$ fails to satisfy condition (†) by $\sum_{j=1}^{d-r+1} \frac{\epsilon}{d} \leq \sum_{j=1}^{d} \frac{\epsilon}{d} < \epsilon$. Using an additional union bound, we can bound the probability that *some* $\Omega_\tau$ fails to satisfy condition (†) by $(r+1)\epsilon$, as desired. $\square$

All that remains is to show that if an entry of $\mathbf{X}^k$ is observed with probability $p$ as in Theorem 2, then $\mathbf{X}^k$ will be observed on enough entries per column. We show this using a simple Chernoff bound.

***Proof of Theorem 2.*** Let $m$ be the number of observations in a column of $\mathbf{X}^{\mathrm{k}}$. Since an entry of $\mathbf{X}^{\mathrm{k}}$ is observed with probability $\mathrm{p}$, then $\mathbb{E}[m] = \mathrm{dp}$, so using the multiplicative form of the Chernoff bound with $\beta = {}^1/_2$ we get:

$$\mathbb{P}\left( m \le \frac{1}{2}\mathrm{dp} \right) \; = \; \mathbb{P}\Big( m \le (1 - \beta)\mathbb{E}[m] \Big) \; \le \; e^{-\frac{\beta^2}{2}\mathbb{E}[m]} \; = \; e^{-\frac{1}{8}\mathrm{dp}} \le \frac{\epsilon}{\mathrm{d}},$$

where the last inequality follows because $\mathrm{p} \ge \frac{8}{\mathrm{d}}\log\frac{\mathrm{d}}{\epsilon}$ by assumption. This shows that with probability $\ge 1 - \frac{\epsilon}{\mathrm{d}}$, a column in $\mathbf{X}^{\mathrm{k}}$ will have at least $\frac{\mathrm{dp}}{2} = \mathrm{m}$ observations, with $\mathrm{m}$ as in (3). Using a union bound on $(\mathrm{r}+1)\mathrm{d}$ columns, we conclude that with probability $\ge 1 - (\mathrm{r}+1)\epsilon$, at least $(\mathrm{r}+1)\mathrm{d}$ columns of $\mathbf{X}^{\mathrm{k}}$ will have $\mathrm{m}$ or more observations, distributed uniformly at random, as required by Lemma 3, which in turn implies that $\mathbf{\Omega}^{\mathrm{k}}$ will satisfy the conditions of Theorem 1 with probability $\ge 1 - 2(\mathrm{r}+1)\epsilon$, as claimed. $\qquad\square$

To guarantee that each $\mathbf{X}^{\mathrm{k}}$ is observed with probability $\mathrm{p}$, we can simply sample uniformly among $\mathbf{X}^1, \dots, \mathbf{X}^{\mathrm{K}}$ with probability $\mathrm{Kp}$, and hence we conclude that the sample complexity of MMC is $\mathcal{O}(\frac{\mathrm{K}}{\mathrm{d}}\max\{\mathrm{r}, \log\mathrm{d}\})$, as claimed.

**Remark 3.** *Notice that we cannot apply Lemma 3 directly instead of Theorem 2, because if we sample* $\mathrm{m}$ *entries selected uniformly at random from each column of* $\mathbf{X}^{\mathrm{k}}$*, there could be* collisions *between multiple matrices in the mixture, which we do not allow, because that would imply observing two values for the same entry in* $\mathbf{X}_{\mathbf{\Omega}}$*.*

## C   More about our Assumptions

Essentially, **A2** requires that $\mathbf{X}$ is a generic mixture of low-rank matrices. There are several equivalent ways to interpret **A1**. For instance, **A2** requires that the columns in $\mathbf{X}^{\mathrm{k}}$ are drawn independently according to an absolutely continuous distribution with respect to the Lebesgue measure on an r-dimensional subspace in general position. Alternatively, recall that every rank-r matrix $\mathbf{X}^{\mathrm{k}} \in \mathbb{R}^{\mathrm{d}\times\mathrm{n}}$ can be expressed as $\mathbf{U}^{\mathrm{k}}\mathbf{\Theta}^{\mathrm{k}}$, where $\mathbf{U}^{\mathrm{k}} \in \mathbb{R}^{\mathrm{d}\times\mathrm{r}}$ and $\mathbf{\Theta} \in \mathbb{R}^{\mathrm{r}\times\mathrm{n}}$. **A2** equivalently requires that the entries in $\mathbf{U}^{\mathrm{k}}$ and $\mathbf{\Theta}^{\mathrm{k}}$ are drawn independently according to an absolutely continuous distribution with respect to the Lebesgue measure on $\mathbb{R}$.

**A2** discards pathological cases, like matrices with identical columns or exact-zero entries, which appear with zero-probability under **A2**. For instance, backgrounds in natural images can be highly structured but are not perfectly constant, as there is always some degree of natural variation that is reasonably modeled by an absolutely continuous (but possibly highly inhomogeneous) distribution. For example, the sky in a natural image might be strongly biased towards blue values, but each sky pixel will have at least small variations that will make the sky not perfectly constant blue. So while these are structured images, these variations make them generic enough so that our theoretical results are applicable.

Furthermore, since absolutely continuous distributions may be strongly inhomogeneous, they can be used to represent highly coherent matrices (that is, matrices whose underlying subspace is highly aligned with the canonical axes). Typical completion theory [1, 2, 5–20, 35, 36] cannot handle some of the highly coherent cases that our new theory covers.

However, we point out that **A2** does not imply coherence nor vice-versa. For example, coherence assumptions indeed allow some identical columns, or exact-zero entries. However, they rule-out cases that our theory allows. For example, consider a case where a few rows of $\mathbf{U}^{\mathrm{k}}$ are drawn i.i.d. $\mathcal{N}(0, \sigma_1{}^2)$ and many rows of $\mathbf{U}^{\mathrm{k}}$ are drawn i.i.d. $\mathcal{N}(0, \sigma_2{}^2)$, with $\sigma_1 \gg \sigma_2$. This is a good model for some microscopy and astronomical applications that have a few high-intensity pixels, and many low-intensity pixels. Such $\mathbf{U}^{\mathrm{k}}$ would yield a highly coherent matrix, which typical theory and algorithms cannot handle, while ours can. To sum up, our assumptions are different, not stronger nor weaker than the usual coherence assumptions [1, 2, 5–20, 35, 36], and we believe they are also more reasonable in many practical applications.

## D   Fine Tuning AMMC

Section 5 presents our alternating algorithm for MMC, summarized in Algorithm 1 below. Like other mixture problems, MMC is highly non-convex, and can be quite challenging in practice. In fact, to

date, there exist no provable practical algorithms for even the simplest mixture problems. Arguably the most common approach is to use alternating EM-type algorithms [16–19, 67–69], which can only be guaranteed to converge to a local optimum, but perform well in practice. Like these algorithms, AMMC also suffers from local minima. Consequently, its performance depends on initialization. In similar classification problems, it is usually convenient to initialize *centers* as *far* as possible. In our case, the centers are the subspaces containing the columns of the matrices in the mixture. Following these ideas, we initialize AMMC with random subspaces as orthogonal as possible.

In addition to initialization, AMMC can be further tailored to specific settings (e.g., **noise**) by making small adaptations. For example, suppose instead of $\mathbf{X_\Omega}$ we observe

$$\mathbf{X_\Omega} \ + \ \mathbf{Z_\Omega},$$

where $\mathbf{Z}$ represents a noise matrix with zero-mean and variance $\sigma^2$. Then, in step 4 of AMMC we can keep erasing entries of $\boldsymbol{\omega}$ until all the entries in $\mathbf{x}_{\boldsymbol{v}^k}$ are within $\sigma^2$ from $\hat{\mathbf{U}}^k$. Alternatively, one can keep in $\boldsymbol{v}^k$ only the $m$ entries of $\boldsymbol{\omega}$ indicating the entries of $\mathbf{x}_{\boldsymbol{\omega}}$ that are *most* likely to correspond to $\mathbf{X}^k$, where $m$ is a tuning parameter.

Similarly, when clustering in step 7, we can keep in $\hat{\boldsymbol{\Omega}}^k$ only the entries of $\mathbf{X_\Omega}$ that are within $\sigma^2$ from $\hat{\mathbf{X}}^k$. Alternatively, we can keep in each $\hat{\boldsymbol{\Omega}}^k$ only the $M$ entries corresponding to the entries of $\mathbf{X_\Omega}$ that are *most* likely to correspond to $\hat{\mathbf{X}}^k$, where $M$ is a tuning parameter that works as proxy of the noise. At the end of the procedure, the entries that not assigned to any $\hat{\boldsymbol{\Omega}}^k$ can be considered outliers, thus providing a robust version of MMC. In fact, this is precisely the approach that we use in our background segmentation experiments in section 6.

Finally, if there is some side information about $\mathbf{X}^k$, it may be beneficial to use a particular LRMC algorithm in step 8 of AMMC. For example, a two-phase sampling procedure [14] may be better if $\mathbf{X}^k$ is coherent. On the other hand, the inexact augmented lagrange multiplier method for LRMC [35, 36] is faster. Iterative hard singular value thresholding [13] is easily implemented and often has similar performance as others [3]. Soft singular value thresholding [6] is better understood and has stronger theoretical guarantees. There are many other methods for LRMC, like OptSpace [7], GROUSE [8], FPCA [10], alternating minimization [16], and LMaFit [11, 12], to name a few. Depending on $\mathbf{X}^k$, it may be better to use one LRMC method or an other in step 8 of AMMC.

---

**Algorithm 1:** Alternating Mixture Matrix Completion (AMMC).

---

1. **input:** Partially observed data matrix $\mathbf{X_\Omega}$.
2. **initialize:** Guess $\hat{\mathbf{U}}^1, \ldots, \hat{\mathbf{U}}^K \in \mathbb{R}^{d \times r}$.

**repeat**

  **CLUSTER:**

  **for** $j = 1, \ldots, n$, and $k = 1, \ldots, K$ **do**

    3. $\mathbf{x}_{\boldsymbol{\omega}} = j^{\text{th}}$ column of $\mathbf{X_\Omega}$.

    4. *Erase* entries from $\boldsymbol{\omega}$ to obtain $\boldsymbol{v}^k \subset \boldsymbol{\omega}$ indicating entries likely to correspond to $\mathbf{X}^k$.

    5. Estimate coefficient of $j^{\text{th}}$ column of $\mathbf{X}^k$:

$$\hat{\boldsymbol{\theta}}^k \ = \ (\hat{\mathbf{U}}_{\boldsymbol{v}^k}^{k\mathsf{T}} \hat{\mathbf{U}}_{\boldsymbol{v}^k}^k)^{-1} \hat{\mathbf{U}}_{\boldsymbol{v}^k}^{k\mathsf{T}} \mathbf{x}_{\boldsymbol{v}^k}.$$

    6. The $j^{\text{th}}$ column of $\hat{\mathbf{X}}_{\boldsymbol{\Omega}}^k$ is given by $\hat{\mathbf{x}}_{\boldsymbol{\omega}}^k \ = \ \hat{\mathbf{U}}_{\boldsymbol{\omega}}^k \hat{\boldsymbol{\theta}}^k$.

  **end for**

  7. Cluster the entries of $\mathbf{X_\Omega}$ according to their closest match among $\hat{\mathbf{X}}_{\boldsymbol{\Omega}}^1, \ldots, \hat{\mathbf{X}}_{\boldsymbol{\Omega}}^K$ to produce $\hat{\boldsymbol{\Omega}}^1, \ldots, \hat{\boldsymbol{\Omega}}^K$.

  **COMPLETE:**

  **for** $k = 1, \ldots, K$ **do**

    8. Complete $\mathbf{X}_{\hat{\boldsymbol{\Omega}}^k}$ using LRMC to obtain $\hat{\mathbf{X}}^k$.

    9. $\hat{\mathbf{U}}^k = $ leading $r$ singular vectors of $\hat{\mathbf{X}}^k$.

  **end for**

**until** convergence.

10. **output:** Completed matrices $\hat{\mathbf{X}}^1, \ldots, \hat{\mathbf{X}}^K$.

---

# E   More Real Data Results

In Section 6 we gave one example of two images from the Yale B [64] dataset being reconstructed from a single mixture. Figure 4 shows more results. Section 6 also shows the segmented foreground of a video frame from the. Figure 5 shows more results from the Wallflower [65] and the I2R [66] datasets.

Figure 4: **Top:** Mixture matrix $\mathbf{X}$, containing pixels from two face images. **Bottom 2:** Low-rank matrices $\hat{\mathbf{X}}^1$ and $\hat{\mathbf{X}}^2$ recovered from $\mathbf{X}$.

Figure 5: Video frames segmented into background and foreground using Robust PCA (displaying the best results amongst [35–39]) and AMMC.