[Reviews · NeurIPS 2018]

Reviewer 1



The paper considers an interesting problem that generalizes both subspace clustering and matrix completion, and provides some sufficient conditions for recoverability. The setting is the following: we are given a matrix A that is a "mixture" of K low-rank matrices, and the goal is to recover them. By a mixture, we mean that every non-zero entry of A is equal to the entry of one of the low-rank matrices. The paper presents a (somewhat strong) sufficient condition under which the original matrix can be recovered in an information theoretic sense. The problem turns out to be fairly difficult, so some such assumptions are necessary for unique recovery. The condition has the feel of Hall's condition for matching; it is also shown to hold w.h.p. in a natural probabilistic model (akin to the one used in "regular" matrix completion, K=1). Overall I find the result interesting. The condition is one that is combinatorial, and it seems unlikely that one can verify if a set of matrices \Omega satisfy the condition in polynomial time. Also, the paper presents some heuristics for the problem, which resembles the K-SVD algorithm for sparse coding, and at an even higher level, Lloyd's algorithm. It's a natural heuristic, and authors show that it works fairly well in applications. Overall, the paper has some interesting ideas about proving unique recovery for a pretty difficult matrix problem. I recommend acceptance.

Reviewer 2



This paper presents mixture matrix completion (MMC) as a novel machine learning tool for learning from low-rank and incomplete data. MMC is a problem that is similar to the problem of subspace clustering with missing data, but is more difficult. Specifically, in MMC the data is assumed to lie in a union of (unknown) low-dimensional subspaces, but the data is not fully observed: only a few entries of each data point are observed, and (unlike subspace clustering with missing data) there is no information as to which entries correspond to the same point. Therefore, one would need to estimate the assignment of entries to data points, the assignment of data points to subspaces, the missing entries, and the subspaces altogether. The major contribution of this paper is the introduction of the MMC problem, a theoretical analysis for when the problem of MMC is well-defined, and an alternating estimation algorithm for solving the MMC problem. Strengths: - The paper presents a new machine learning problem formulation that seems natural for addressing practical tasks such as background segmentation. It also has a nice preliminary study of this problem in terms of atheoretical analysis of the identifiability problem, a simple practical algorithm, and experiments on real and synthetic data. - The paper is well written and the logic flow is clear. Weaknesses: - My major concern with the paper is that the theoretical results seem to be stated in vague terms and I don't fully understand them. In Theorem 1, what does it mean to say that Omega^k "contains" disjoint matrices Omega_tau? Does it mean that Omega^k is a stack of matrices Omega_tau column-wise? Also, what does it mean that it is "possible" to perfectly recover all columns of X^k? Does it mean that the subspaces and the missing entries can be uniquely determined from the observed data? In addition, how is this result compared with the deterministic result for the problem of subspace clustering with missing entries in [4]? Overall this is a nice paper that brings up a new problem formulation into attention. The study of this problem is still preliminary, though, as one can clearly see that there is no very successful application of the proposed method yet. The experiment on face clustering has a very unrealistic test scenario, and the experiment for background segmentation does not generate results as good as classical robust PCA. Nonetheless, addressing these challenges could be the topic of future study. My reason for not having a higher rating is that I cannot fully appreciate the theoretical studies as mentioned above. Additional comments: - It appears that the "cluster" step in the proposed algorithm is very complicated. I'm wondering if this step can be decomposed into the following substeps to make it easier to explain: the estimation of Omega_k is composed of two separate tasks of 1) clustering the entries in each column to different data points, and 2) the assigning data points extracted from all columns to multiple subspaces. In fact, once one have solved task 1) above then the problem reduces to subspace clustering with missing data, which could be solvedby alternating between matrix completion and subspace clustering. - Since the overall problem is nonconvex, initialization is expected to be very important for the algorithm to achieve good performance. Can the authors comment on how their algorithm is initialized on real data experiments? Response to rebuttals: The updated statement of Theorem 1 in the rebuttal is now much better in terms of clarity and it appears to be very necessary to incorporate it into the final version. Also, given that the conditions in Theorem 1 and 2 are very much similar to those in [4] for a related problem, a more detailed discussion of their connections will help understand the merits of these results. I maintain my overall rating as above and recommend a weak acceptance for this work.

Reviewer 3



This paper proposes a new variation of the matrix completion problem, called mixture matrix completion (MMC), where each entry of the matrix is drawn from one of a few low-rank matrices, rather than the same low-rank matrix. The proposed problem seems to be valid with motivations from a few applications. This paper makes two contributions: 1) an information-theoretical lower bound on the sample complexity; and 2) a heuristic algorithm to solve the MMC problem based on alternating minimization. The paper is written clearly with sufficient backgrounds information and provides extensive numerical experiments. The theoretical results of the paper are rather weak. The info-theoretical bound is straightforward and directly follows from previous studies on the matrix completion problem in [3], and [4] combined with a combinatorial enumeration. -Moreover, the statement of Theorem 1 is a bit hard to follow, and in some parts the meaning is unclear. For example, it is not clear how one can use Theorem 1 to verify if a given pattern can be used to solve the MMC problem in a computational-efficient way. Does "it is possible to" mean there exist an algorithm to recover ..? -In Theorem 2, it requires the number of columns needs to be about r times larger than the number of rows, which is a strong assumption. For example, this eliminates the applicability of this result on square matrices. Is this requirement always needed? For the alternating algorithm (AMMC), the main issues are 1) how to select the initialization in a data-driven manner or adaptively; I haven't found any discussions on it; 2) an analysis of the computational complexity of the proposed AMMC algorithm. A major issue in the real-data experiment is that the AMMC algorithm uses down-sampled data, and for such, many details in the images for the segmentation experiments are lost. For example, the background trees may be quite smoothed after down sampling, and much easier to separate. Therefore, the performance improvement showed in Figure 3 may not come directly from the new algorithm but an artifact of downsampling. In Figure 5 of the supplementary material, row 3 and row 7, there show more people than appeared in the original frame, can you explain why? In summary, the paper proposed an interesting problem (MMC) to study, but the results are rather immature to be published in its current form. Small typos: -line 168, the word "entries" appeared twice -line 170, the word "and" appeared twice update: I have read the authors' rebuttals and below are updates of my review. First, thanks for the authors' clarifications of many aspects of the work which helped my understanding. My main concerns are: - clarity as well as the novelty of the theory: The theorems of this paper are built heavily on existing results in [3] and [4] and the additional arguments appear to me as incremental. Furthermore, the applicability of Theorem 2 to square matrices is still unclear form the rebuttal; the authors claim it is applicable but it is not clear how since there is an assumption in Theorem 2 that explicitly prevents it from being applied. It seems a lot of work are needed to make the statements of the theorems clear (I do appreciate the authors' efforts in the rebuttal to make them more clear than the submitted version already); - the authors' acknowledged the unfairness in the comparison between RPCA with full data and their algorithm with subsampled data and updated the simulations. I am not sure why the tree in the backgrounds, while stay blurred in the reconstruction of the backgrounds, its residual didn't show up in the foreground in the authors' algorithm.